# A Study on the Mechanism of Digital Technology's Impact on the Green Transformation of Enterprises: Based on the Theory of Planned Behavior Approach

Yi Gao [1] and Yinkai Tang [2,*]

1   School of Economics and Management, Xi'an University of Post and Telecommunications, Xi'an 710061, China; gaoyi@xupt.edu.cn
2   School of Modern Post, Xi'an University of Posts and Telecommunications, Xi'an 710061, China
*   Correspondence: tangyinkai@stu.xupt.edu.cn

**Abstract:** With the rapid development of the digital economy and the continuous improvement of the digital capabilities of enterprises, relying on digital technology (DT) to achieve green transformation (GT) has become the future development direction of enterprises. Based on the theory of planned behavior (TPB), this paper constructs a theoretical model of DT to determine the impact mechanism of corporate GT and empirically tests the research model using a structural equation model (SEM). The analysis of microdata from 406 manufacturing firms in China shows that DT has a positive contribution to corporate GT. DT mainly affects the intention of enterprises to pursue GT indirectly by influencing the perceived behavioral control for GT and thus, ultimately, the GT of enterprises. This paper reveals the model and mechanism of corporate GT through DT, which has important implications for relevant theoretical research and policy formulation.

**Keywords:** digital technology; green transformation; theory of planned behavior (TPB); structural equation model (SEM)

## 1. Introduction

Since reform and opening up, China's rapid economic development has come at a huge cost in terms of resources and the environment. Striking a balance between economic development and environmental protection has become an urgent and realistic problem to be solved. Strengthening the construction of ecological civilization and promoting comprehensive GT is the only way to achieve high-quality development. In recent years, the Chinese government has attached great importance to the green development of the economy and has introduced many policies to actively guide the GT of the real economy. As a micro-level subject of economic development and environmental protection, the GT of enterprises is not only an effective starting point for a country to promote the GT of its economy but also an important object of attention in formulating green economic policies [1]. Therefore, to achieve the GT of China's economy, it is necessary to promote the GT of enterprises and achieve the unification of economic, social, and environmental benefits in green development.

With the development of DT such as big data, cloud computing, artificial intelligence, and blockchain, the digital economy, with data as a key factor of production, has ushered in new development opportunities. The China Academy of ICT's China Digital Economy Development Report (2023) shows that in 2022, China's digital economy reached 50.2 trillion yuan, a nominal growth of 10.3% year-on-year, which has been significantly higher than the nominal GDP growth rate for 11 consecutive years, and the digital economy accounted for 41.5% of GDP, equivalent to the share of secondary industry in the national economy. In the era of the digital economy, the pace of enterprise digitalization has accelerated, and the promotion of intelligent manufacturing, application of new DT, adoption of Internet

business models, and building of modern information systems in production and operation are profoundly changing the production methods, organizational forms, marketing models, management models, and business strategies of enterprises [2]. The digitalization of enterprises is essentially a dynamic process of introducing and applying DT by investing large amounts of capital and human resources to cope with uncertain technological changes and market fluctuations [3]. Promoting the GT of enterprises is a key task in establishing and improving an economic system for green, low-carbon, and circular development, while digitalization has become a major driver of transformation and is considered to have great potential for promoting structural reform on the supply side and achieving sustainable enterprise development [4]. In other words, in the era of the digital economy, the GT of enterprises is no longer simply a matter of technological upgrading, as it has gradually become a strategic choice affecting the green and sustainable development of enterprises [5].

At this stage, the level of initiative for GT in Chinese manufacturing enterprises is poor, and the drivers of transformation mainly rely on external pressure to push enterprises to achieve GT [6]. At the same time, most of the existing empirical studies on GT focus on the macro level, such as cities and industries, while ignoring the behavioral processes in microenterprises' GT [7]. There is less literature on research from the internal perspective of microenterprises, which makes enterprises' green transformation behavior lack systematic theoretical guidance. In the context of China's vigorous digital transformation, social and environmental values are increasingly becoming important elements in building sustainable competitive advantage for companies [8]. This is because digitalization has greatly increased the social transparency of enterprises, information asymmetry has been significantly alleviated, and stakeholders have higher expectations for enterprises to fulfill their environmental responsibilities, which will drive them to take the initiative to make a GT. However, there are currently two main problems with the GT of enterprises. One is the lack of motivation. According to economic theory, the sole purpose of enterprises is to maximize profits. When considering profit, enterprises generally believe that environmental investment will take up some of their productive resources and therefore lack the will to actively invest in environmental protection. Secondly, there is a lack of capacity. The lack of technical and human resources support in the process of GT in enterprises has led to difficulties in identifying environmental problems, information asymmetry, and an insufficient basis for strategic decision making. It has been pointed out that digitalization as part of economic and social development in a wide range of areas will eventually be internalized in the GT of enterprises [9]. Therefore, in the context of the current urgent need to accelerate the development model, the digitalization process of enterprises will provide inspiration for them to further seek green development and adopt GT, which is an important reason for introducing the exogenous variable of digitalization technology in this paper. So, can DT effectively contribute to the GT of enterprises? And how does DT contribute to the GT behavior of enterprises? The innovative exploration of the mechanism of DT's impact on corporate GT at this stage undoubtedly enriches the current immature theoretical system and analytical framework and is of great theoretical and practical significance to the promotion of corporate GT and sustainable development in China.

Based on this, this paper conducts an empirical study on the mechanism of DT's impact on the GT of enterprises in China based on the actual situation of Chinese enterprises, focusing on clarifying the following two questions: (i) Can DT drive enterprises to adopt GT behavior? (ii) What are the intermediate mechanisms of influence of DT on enterprises' GT? The marginal contributions of this paper may lie in the following: First, while most existing studies have focused on a small number of explorations of GT at the macro level, such as cities and industries, this paper takes a micro perspective and directly establishes the association between DT and corporate GT, bridging the research gap in this micro area. Second, based on the TPB, an SEM is used to explore the intermediate transmission mechanism of the impact of DT on the GT of Chinese enterprises, in an attempt to open

the "black box" of the GT process of digitally empowered enterprises and expand the theoretical logic of the non-economic value creation of DT.

## 2. Literature Review

### 2.1. The Meaning and Impact of DT

DTs are playing an increasingly important role in driving the global economy, impacting society, businesses, and people's lives [10]. These technologies include but are not limited to big data, artificial intelligence, cloud computing, blockchain, and the Internet of Things [11]. Digital transformation has triggered the transformation of new business models [12] based on new logic and ideas to help companies upgrade their existing technologies, products, and business processes to become more competitive in the marketplace [13].

With the increasing maturity and widespread use of DT, the academic understanding of DT mainly covers three levels: macro, meso, and micro. At the macro level, DT uses data as a key factor of production to promote the reallocation of existing factors of production and carry out a series of economic activities on a network platform [14], thus triggering fundamental changes in social production methods and economic structures and giving rise to a new form of economic development, the digital economy [15]. The deepening application of DT has facilitated the development of the digital economy, while the extensive penetration of new technologies and new economic forms has provided new impetus for economic growth [16]. At the meso level, DT integrates the data resources of all enterprises in an industry to form a data platform [17], and the platform empowers the digital transformation of the industry by improving operational mechanisms, sharing data resources, and choosing different role points, priorities, and methods to continuously promote the digital transformation of the industry [18]. Most of the micro-level-based DT research is for enterprises, emphasizing the application of DT and the process of organizational change. For example, Meng [19] argues that the essence of enterprise digitalization is the corporate strategic behavior of business organizations using DT, but Vial [20] argues that enterprise digital transformation is a process of organizational change in which companies change their path of value creation through the application of DT, thereby improving internal operational efficiency and organizational performance. Taken together, enterprise digitalization encompasses the application of DT at multiple levels, such as production and sales methods, operational decisions, business models, and even value chain relationships, to create and capture enterprise value.

Based on existing research, DT has been shown to have a profound impact on business activities. Firstly, digitalization of an enterprise can effectively reduce information communication and transaction costs and improve asset utilization, thereby increasing economic efficiency [21]; secondly, digitalization of an enterprise improves managers' business perceptions and decision-making capabilities by changing corporate strategies, business processes, products, and services [22], enabling companies to gain a competitive advantage in an uncertain economic environment [23]; thirdly, digitalization technology empowers companies with innovation resources and innovation dynamics, which can enhance the ability of firms to respond to changes in the external environment [24]. Notably, scholars have also found that DTs have an impact on environmental improvement [25] and that they promote healthy and sustainable business development [26].

In the digital age, barriers to business and social communication are significantly reduced, so companies can quickly and accurately capture consumer demand for corporate environmental responsibility [27], while DT drives the ability of companies to fulfill their environmental responsibilities [28]. Furthermore, Camodeca and Almici [29] found that DT significantly improved CSR and helped companies achieve their sustainability goals. In the context of supply chains, the use of big data analytics by companies can positively moderate the relationship between sustainable supply chain management and organizational performance [30]. With digital transformation opening up new growth opportunities for businesses and corporate environmental management being a key corporate governance

issue, there is an urgent need to clarify how companies can effectively use DT to enhance the effectiveness of corporate environmental management.

*2.2. Research on GT*

The GT of enterprises is generally considered to be the initiative of enterprises to adjust their strategies to reduce resource consumption and change from a wasteful and polluting development model to a green and sustainable development model that conserves resources and protects the environment [31]. GT can effectively help enterprises bring about the value of symbiosis between ecological and economic benefits, ultimately leading to an optimal allocation of resources.

Existing studies have mainly explored the factors influencing the green transition from the perspectives of external and internal factors. From the perspective of external factors, most scholars study the impact of environmental regulations on GT. For example, environmental regulations [32], low-carbon policies [33], environmental taxes [34], and other environmental regulations can effectively restrain the behavior of enterprises, raise their awareness of environmental protection, prompt them to implement green business concepts, and promote their green and sustainable development. At the same time, the government compensates for the high costs and risks associated with GT through government subsidies to motivate enterprises to actively carry out GT [35]. The strong influence of the external environment on the GT of enterprises is also reflected in the pressure from stakeholders to adopt environmentally responsible behavior. Green products can bring private benefits to stakeholders in addition to the public benefits of reducing environmental pollution [36], so the environmental needs of customers, suppliers, and other stakeholders can also influence corporate green behavior [37]. Telecommunications infrastructure development can improve the level of information technology of enterprises, increasing the space for progress and the speed of dissemination of green technologies, thus creating incentives for sustainable development [38]. A sound financial mechanism can provide financing services to enterprises and increase liquidity, thus encouraging innovation [39], so developing green finance can help stimulate enterprises and promote new green projects [40].

As far as internal factors are concerned, scholars have analyzed and discussed them mainly in terms of corporate organizational characteristics. For example, corporate governance structures [41], organizational resources [42], and the environmental awareness of executives [43] can promote active environmental projects internally, shape employees' environmental behavior [44] and help companies achieve their green development goals. In addition, technological innovation is believed to have a driving effect on corporate transformation [45]. Endogenous growth theory suggests that technological progress is an important cause of sustained corporate growth [46], and technological innovation has an important impact on the green development aspect of a company by improving production efficiency and reducing production costs through technological advances [47]. Green technological innovation can reduce environmental pollution, save energy, and achieve green and sustainable development balancing environmental protection and enterprise competitiveness [48], which is a key path to harmonious economic and environmental development.

On the adoption of GT behaviors by enterprises, existing research has focused on two aspects, namely green strategy and green innovation [49]. On the one hand, corporate transformation requires strategic leadership [50], and Sun [51] argues that corporate GT is essentially strategic transformation, while Eric and Olson [52] argue that green strategy fundamentally helps companies make decisions to improve the environment as well as facilitate corporate transformation. On the other hand, green technology innovation enhances the ability of firms to carry out green production, and it is the key to the GT of firms [53]. Xiao [54] found that green innovation capability can effectively improve GT performance after histological analysis of manufacturing GT cases and that technological conditions play a central role. Li [48] further proposed that green innovation is an inexhaustible driving force for the green development of manufacturing enterprises, which not only helps to improve the environmental pollution problems of enterprises but also helps to improve

enterprise performance. Green strategy and green innovation are closely related: to carry out green innovation, enterprises need to adopt green strategy measures for orientation first, and green innovation in turn realizes the value of green strategy [55]. Green strategy and green innovation integrate corporate GT behaviors and together have a positive impact on corporate transformation performance.

## 3. Research Model and Hypotheses

### 3.1. Theory of Planned Behavior

The theory of planned behavior (TPB) argues that the drivers of individuals' behavioral decisions mainly include behavioral attitude variables, subjective normative variables, and perceived behavioral control variables, which together indirectly influence behavior through behavioral intentions [56]. This theory provides a valid pathway for factors influencing environmental behavior and green development in companies [57], which has strong explanatory power. The introduction of "DT" extends the traditional TPB model [56] to investigate the mechanisms and pathways through which DT influences the GT of enterprises. As the TPB model is less commonly used in the field of GT behavior research, this part of the study should first determine how well the model fits the data related to GT behavior.

### 3.2. DT

According to environmental adaptation theory, enterprises need to respond quickly to the changing external environment [58]. The digital development environment formed against the backdrop of the digital economy is a brand-new external environment faced by enterprises. With the advent of the digital economy, digitalization is becoming the main driver for enterprises to carry out transformation and upgrading [59]. According to the resource-based view, digitalization can help enterprises search for more favorable information and improve their ability to analyze and process information [60]. DT gives full play to the advantages of resource sharing and efficient information circulation, helping enterprises make scientific decisions that are conducive to the joint development of economic benefits and environmental protection, improving their operational efficiency, and reducing the cost of carrying out GT [48]. In addition, enterprises can adapt to a rapidly changing environment and make dynamic adjustments. In the process of scaling up, enterprises need dynamic capabilities to create and maintain a competitive advantage over other enterprises [11]. Firms can maintain sustainability and improve innovation efficiency through digital transformation [61]. Digitalization is the key to gaining a new competitive advantage for traditional firms and is a major source of green innovation capabilities. The high internal and external costs for companies to undertake GT, as well as the lack of corporate capacity, ultimately lead to a low intention to pursue GT. Determining how to maximize the motivation of enterprises to pursue GT through DT is, then, the most important purpose of introducing DT as an exogenous variable into the TPB. Based on the above discussions, this paper puts forward the following hypotheses:

**Hypothesis 1 (H1).** *DT positively impacts the attitude that digitalization drives GT.*

**Hypothesis 2 (H2).** *DT positively impacts the subjective norm for GT.*

**Hypothesis 3 (H3).** *DT positively impacts the perceived behavioral control for GT.*

### 3.3. Attitude (Digitalization Drives GT)

Previous studies have shown that the adoption of GT behavior is directly related to changes in the environmental perceptions of corporate managers, but environmental investment and the cost of transformation are the main barriers to GT in companies. It is difficult to generate significant economic benefits in the short term, but in the long term, it can be beneficial for companies to gain a competitive advantage [62]. Like the GT of development approach, improving digitalization can also benefit companies in the long

term. The expected benefits created for companies by promoting a GT through increased digitalization and the reduced internal and external costs to the company can give managers a positive estimate of the results and value of the GT. Thus, in conclusion, the following hypothesis was developed:

**Hypothesis 4 (H4).** *The attitude that digitalization drives GT positively impacts the intention to pursue GT.*

### 3.4. Subjective Norm for GT

Enterprises adjust dynamically to changes in the internal and external environment, taking proactive actions and making responses. Firstly, companies with corporate environmental responsibility are more likely to apply green, low-carbon principles and values in their strategy development and implementation and to discipline their environmental investment behavior [63]. Environmental policies and the relationship between companies and external stakeholders are important influencing factors in companies making a green transition. Companies will implement environmental strategies to meet policy regulations and changes in consumer demand, which will influence green transition behavior [64]. In addition, GT by companies in the same industry or by reference companies in the development process brings about a demonstration effect. Even if a company does not participate in the GT process itself, the performance of competing companies through the GT process and the lessons learned through the demonstration effect will have a positive impact on their intention to make a GT. Based on the above discussion, we propose the following hypothesis:

**Hypothesis 5 (H5).** *The subjective norm for GT positively impacts the intention to pursue GT.*

### 3.5. Perceived Behavioral Control for GT

The resource element of an enterprise is the basis of GT [65]. When considering whether a company should adopt a GT, managers look at the multiple types of resources such as talent, capital, and technology that the company itself possesses and acquires, and they make decisions based on their capabilities and realities [66]. According to RBV, organizational resources enhance corporate green manufacturing capabilities [67], which can enable companies to achieve operational and environmental performance excellence [68]. When the knowledge and skills acquired by the company are enhanced, the managers' willingness to pursue GT increases. Based on the above analysis, the following hypothesis is formulated in this paper:

**Hypothesis 6 (H6).** *The perceived behavioral control for GT positively impacts the intention to pursue GT.*

### 3.6. Intention to Pursue GT

Behavioral intentions explain the degree of effort an actor is willing to put into an action. In general, the more positive the behavioral attitude of the actor, the greater the support from the external environment, and the stronger the perceived behavioral control, the greater the probability that a target behavior will be performed [69]. Intention to pursue GT is a subjective measure of a company's willingness to adopt GT behaviors, reflecting the level of acceptance and approval of GT behaviors. It has been suggested that a company's green innovation intentions are a motivational factor influencing green innovation behavioral responses [70] and that intentions have a direct positive impact on behavior [71]. Therefore, the stronger a company's intention to pursue GT is, the more likely it is to adopt GT behavior to cope with the dynamic and changing internal and external environment of the company. Thus, we provide the following hypothesis:

**Hypothesis 7 (H7).** *The intention to pursue GT positively impacts the GT of enterprises.*

### 3.7. The Mediating Role

This paper proposes that the impact of DT on enterprise GT resulting from increasing the digitalization of manufacturing enterprises is based on the following three main aspects. Firstly, based on the energy-saving and emission reduction effects on the production and consumption sides, manufacturing enterprises can achieve green production by improving their processes and optimizing their production methods in a timely manner. With the help of DT such as the Internet, big data, and cloud computing, enterprise digitalization can achieve effective integration of man and machine, quickly capture the environmental impact of production, and collect consumers' environmental demands, which helps enterprises adjust their production status, make precise decisions, and reduce the risk of product development deviating from or lagging behind market changes [72]. Second, based on the focus theory of normative conduct, manufacturing enterprises can promptly scan their external environment, identify possible opportunities and threats, integrate data information, form digital thinking, and use digital information to accurately predict and quickly respond to external green and low-carbon development needs. Digital-aware behavioral norms can promote environmentally friendly behaviors [73], and manufacturing companies can seize the opportunities of the "green and low-carbon" and "digital" era to accelerate the integration of internal and external knowledge, thus promoting GT. Finally, DT promotes GT through cost effects and empowerment effects. On the one hand, the digitalization of enterprises helps reduce the cost of emission reduction and innovation, while lowering the threshold and risk of R&D innovation and low-carbon production. On the other hand, for enterprises' green product innovation and low-carbon production, digital transformation is conducive to enhancing enterprises' information acquisition and processing capabilities, helping them to identify the value of external information, optimize their business and strategies, clarify the direction of product and service enhancement, expand the width of product lines, and formulate green innovative product development plans that focus on users' individual needs to capture development opportunities in potential and marginal markets. This will enable them to optimize the allocation of production factors [74]. For enterprises in the process of continuously improving the efficiency of resource allocation, the green growth of enterprises stimulates the green development momentum of DT. As a comprehensive analysis of the above, this paper puts forward the following hypotheses:

**Hypothesis 8 (H8).** *The variable "DT" indirectly and positively impacts the intention to pursue GT by impacting the attitude that digitalization drives GT and ultimately positively impacts actual behavior towards the GT of enterprises.*

**Hypothesis 9 (H9).** *The variable "DT" indirectly and positively impacts the intention to pursue GT by impacting the subjective norms for GT and ultimately positively impacts actual behavior towards the GT of enterprises.*

**Hypothesis 10 (H10).** *The variable "DT" indirectly and positively impacts the intention to pursue GT by impacting the perceived behavioral control for GT and ultimately positively impacts actual behavior towards the GT of enterprises.*

### 3.8. Theoretical Model

Based on the literature review and research hypotheses, a theoretical model is constructed in which both internal and external factors influence the impact mechanism of GT, as shown in Figure 1.

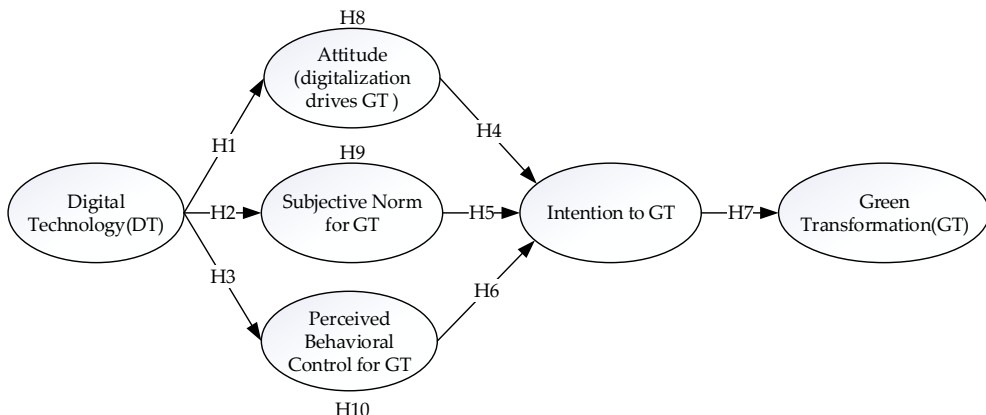

**Figure 1.** The theoretical model of the mechanism of DT's impact on the GT of enterprises.

Based on the connotation of DT and the characteristics of GT behavior, this paper defines behavioral attitude as the subjective attitude of enterprises towards GT driven by digitalization from a micro perspective. It defines the subjective norm as the extent to which internal and external social pressure drives enterprises' GT behavior. In addition, based on the definition of perceived behavioral control variables, the perceived behavioral control variables are defined in this study as the extent to which enterprises can carry out GT.

## 4. Research Design

### 4.1. Variable Measurement

#### 4.1.1. DT

The digitalization of an enterprise consists in the changes and transformations that enterprises can make through DT such as big data, cloud computing, and social platforms. Drawing on the studies of Chi [75] and Hu [76] and related scales of DT abroad [20,77], four indicators of DTs were identified, including the operation, integration, transition, and diffusion of enterprises based on DTs, which were represented by DT1, DT2, DT3, and DT4, respectively.

#### 4.1.2. Attitude (Digitalization Drives GT)

Digitalization on the production side brings new changes in production processes, decision making, and environmental monitoring to enterprises and becomes the basis for GT [78,79]. At the same time, digitalization on the service side enables enterprises to provide targeted advice and services to customers and move towards intelligence, proactivity, and personalization, helping enterprises to achieve their GT through demand-side-driven green development [9]. Based on this, the behavioral attitude that digitalization drives GT, represented by ATT, is measured from the production side and the service side in 2 dimensions, and 5 question items are set, which are represented by ATT1, ATT2, ATT3, ATT4, and ATT5, respectively.

#### 4.1.3. Subjective Norm for GT

This study defines subjective norms as the extent to which internal and external social pressures drive corporate GT behavior. This category was developed based on the research theory of Cialdini [80], which argues that subjective norms consist of personal, exemplary, and directive norms. Personal norms are internal moral pressures felt by individuals to perform or not perform a certain behavior [81], and for green transition personal norms, which indicate the corporate social responsibility felt by companies when making decisions, this paper draws on a scale developed by Chang [82]. Exemplary norms are subjective normative perceptions of specific behaviors that are modeled by outsiders through their behaviors and thus motivate individuals to learn and imitate them [83], and they are represented here by peer pressure to compete. Directive norms are mainly derived from individual constraints on green transition behavior from outside the firm and are measured

here by government regulation and market incentives. Based on this, the subjective norms for GT, represented by SN, are divided into 3 dimensions, with a total of 5 questions, which are represented by SN1, SN2, SN3, SN4, and SN5, respectively.

### 4.1.4. Perceived Behavioral Control for GT

Perceived behavioral control for GT was represented by PBC. According to Ajzen and Fishbein's theory [84], perceived behavioral control is a combination of internal control beliefs (including the individual's emotions, abilities, strengths, weaknesses, etc.) and external control beliefs (including the external environment, information, opportunities, obstacles, etc.). In this questionnaire, external control beliefs reflect the economic indicators and technical difficulties in the industry, with two questions. Internal control beliefs, on the other hand, reflect the economic strengths and technical capabilities of individual firms for GT and are combined with studies by Lin [85] and Cabral [86] to create a total of four questions. Thus, the scale consists of 2 dimensions, external control beliefs and internal control beliefs, and 6 items, which are represented by PBC1, PBC2, PBC3, PBC4, PBC5, and PBC6, respectively.

### 4.1.5. Intention to Pursue GT

According to Gollwitzer [87], behavioral intentions are divided into two stages, the first being the formation of motivation and the second being the formation of plans. In this questionnaire, the intention to pursue GT, represented by INT, was measured by setting variables according to each of these two stages. Therefore, based on Antón's [88] study, four questions were set to cover the intensity of GT intentions in the motivation formation stage and the intensity of GT intentions in the plan formation stage, which are represented by INT1, INT2, INT3, and INT4, respectively.

### 4.1.6. GT

This questionnaire measures the GT of enterprises over time as an approximate proxy for the actual GT behavior of enterprises after they have been surveyed. Based on the research of Yang [89] and Chen [90] and the scale developed by Hou [91] and Chan [92], this questionnaire contains two dimensions, green strategy and green innovation, with six questions in total, which are represented by GT1, GT2, GT3, GT4, GT5, and GT6, respectively. Green strategy refers to an enterprise's strategy of adopting "clean technology" to improve or change its production and operation activities to achieve the strategic goal of saving resources and protecting the environment [93]. Green innovation integrates green development and innovation drives, and it is a key choice for enterprises to achieve GT and improve performance [94].

### 4.2. Questionnaire Design and Data Collection

This section shows that in the process of designing this questionnaire, to improve its reliability and validity as much as possible, this study drew on some of the methods and rules of the literature on questionnaire design and modified them to suit the characteristics of research on DT and GT. In this questionnaire, at least three observed variables were set for each latent variable [95]. In terms of scale selection, all relevant latent variables were measured using a Likert 7-point scale [96], with scores ranked from smallest to largest, indicating low to high levels of endorsement, specifically 1 (completely disagree), 2 (disagree), 3 (slightly disagree), 4 (unsure), 5 (slightly agree), 6 (agree), and 7 (completely agree).

A random sampling method was used to select the sample, and the data were obtained through an online and offline questionnaire, with one questionnaire being completed per enterprise. The questionnaire briefly describes the study and asks to be completed by a senior manager in the production department or R&D department within the firm. In total, 500 questionnaires were randomly distributed among manufacturing enterprises across China from January to April 2023. Of these, 437 questionnaires were returned, 406 of which were valid, for a valid return rate of 81.2%. The enterprise characteristics are shown in

Table 1. The questionnaire consisted of 30 questions based on six latent variables: "DT", "Attitude (digitalization drives GT)", "Subjective Norms for GT", "Perceived Behavioral Control for GT", "Intention to GT" and "GT" (as shown in Table 2). All the questions in this questionnaire are declarative, allowing the respondents to choose their level of agreement with the topics in the questionnaire according to the specific situation of their company. The questionnaire was mainly completed by senior and middle managers who are familiar with green development and environmental governance within the company, thus enhancing the objectivity and validity of the data.

**Table 1.** Descriptive statistics of samples.

| Item | Sample Characteristics | Number of Samples | Percentage (%) |
|---|---|---|---|
| Enterprise types | Traditional manufacturing enterprises | 134 | 33.0 |
| | High-tech manufacturing enterprises | 213 | 52.5 |
| | Others | 59 | 14.5 |
| Enterprise nature | State-owned enterprises | 181 | 44.6 |
| | Private enterprise | 174 | 42.9 |
| | Foreign-funded enterprises | 26 | 6.4 |
| | Others | 25 | 6.2 |
| Enterprise size (persons) | <100 | 28 | 6.9 |
| | 100–500 | 173 | 42.6 |
| | 500–1000 | 184 | 45.3 |
| | >1000 | 21 | 5.2 |
| Enterprise age (year) | <2 | 28 | 6.9 |
| | 2–5 | 170 | 41.9 |
| | 5–10 | 173 | 42.6 |
| | >10 | 35 | 8.6 |

**Table 2.** Definitions and summary statistics of the variables in the questionnaire.

| Latent Variables | Dimensions | Label | Items |
|---|---|---|---|
| Digital Technology(DT) | Operation | DT1 | The company is performing digital-technology-based business processes. |
| | Integration | DT2 | The company is integrating digital technologies to transform our business processes. |
| | Transition | DT3 | The company is shifting its operational management towards the use of digital technologies. |
| | Diffusion | DT4 | The company is willing to put effort into promoting and publicizing digital skills and management knowledge. |
| Attitude (digitalization drives GT) (ATT) | Production side | ATT1 | Digitalization of enterprises can help companies achieve green production. |
| | | ATT2 | Digitalization of the enterprise can provide data to support decisions on green behavior in the company. |
| | | ATT3 | Digitalization of an enterprise can effectively capture the environmental impact of a company's production. |
| | Service side | ATT4 | The company can use its digital platform to collect the environmental needs of consumers. |
| | | ATT5 | The company can enhance its green image through digital services. |
| Subjective Norm for GT(SN) | Personal norm | SN1 | The company has a clear and specific environmental policy. |
| | Exemplary norm | SN2 | Awareness of energy conservation and emission reduction has generally increased in the same industry. |
| | | SN3 | Carrying out green and low-carbon production and operation has become the norm in the industry. |
| | Directive norm | SN4 | National energy-saving standards, relevant policies, and regulations have prompted the company to develop environmentally friendly projects. |
| | | SN5 | Consumer demand for green products has led to the development of environmentally friendly projects. |

**Table 2.** *Cont.*

| Latent Variables | Dimensions | Label | Items |
|---|---|---|---|
| Perceived Behavioral Control for GT(PBC) | External control belief | PBC1 | Energy efficiency and emission reduction technologies are now relatively mature and easy to master. |
| | | PBC2 | Adopting a green transformation is not significantly more costly than a non-green transformation. |
| | Internal control belief | PBC3 | The company has greater access to financial services information and financial products. |
| | | PBC4 | The company can quickly identify its environmental problems and find solutions. |
| | | PBC5 | The company has sufficient resources and manpower to undertake the green transformation. |
| | | PBC6 | Overall, the company has the financial strength and technical requirements to make the green transformation. |
| Intention to Pursue GT (INT) | Formation of motivation | INT1 | The company is willing to carry out pollution control in our production operations. |
| | | INT2 | The company is willing to adopt technologies and equipment related to green transformation. |
| | Formation of plans | INT3 | We will provide a plan to validate the green transformation concept. |
| | | INT4 | The company will organize the exchange of green transformation ideas across all departments. |
| Green Transformation (GT) | Green strategy | GT1 | The company actively monitors pollution emissions and carries out pollution prevention. |
| | | GT2 | The company minimizes the potential harm to the environment during the production of our products. |
| | | GT3 | The company actively introduces clean technologies. |
| | Green innovation | GT4 | The company invests more in research and development of green technologies. |
| | | GT5 | The company actively develops green products. |
| | | GT6 | The company uses greener raw materials as much as possible. |

### 4.3. Structural Equation Modeling

Research in the literature suggests that SEM as a multivariate data analysis tool has strengths in analyzing the relationships between multiple variables. Therefore, this paper uses SEM with observed and latent variables to test theoretical models of corporate GT decision mechanisms [97,98] and to assess the strength of the hypotheses. Schumacker and Lomax [99] found that most SEM studies had sample sizes between 200 and 500. There have also been studies assessing the effect of sample size on SEM results [100,101]. These investigations suggest that a sample of at least 100 cases should be used in latent variable analyses and that fewer than 100 observations result in unreliable estimates of the overall parameters. According to the recommendations of Hair [102], the ratio between observed variables and sample size should be between 1:10 and 1:15 in the process of modeling structural equations. In the model developed in this study, there were a total of 6 structural surfaces and 30 questions, so the 406 samples used in this study met the requirements of the SEM.

The data were analyzed using SPSS 26.0 and AMOS 26.0. The study followed the two-step approach proposed by Anderson and Gerbing [103] for the measurement model and the structural model.

### 5. Results

#### 5.1. Reliability and Validity of the Measurement Model

In this paper, confirmatory factor analysis (CFA) is used to test the reliability and validity of the measurement model [104]. Construct reliability measures the extent to which a construct is free from random error, thereby producing consistent results. According to the general judgment indicators of questionnaire reliability, a Cronbach's alpha coefficient of more than 0.8 indicates good internal consistency; if the Cronbach's alpha coefficient is between 0.7 and 0.8, then internal consistency is good; if the Cronbach's alpha coefficient

is less than 0.7, then internal consistency is poor [105]. The test results show that the Cronbach's alpha coefficient for the overall scale was 0.922, and the Cronbach's alpha coefficients for DT, attitude (digitalization drives GT), subjective norms for GT, perceived behavioral control for GT, intention to pursue GT, and GT were 0.809, 0.853, 0.874, 0.893, 0.801, and 0.879, all of which exceeded 0.8. In addition, this study also used the composite reliability (CR) coefficient for reliability measurement. According to the results in Table 3, the CR values of all the compositional surfaces are greater than 0.7, which means that the measurement model has high compositional reliability [106].

**Table 3.** Results of the reliability and convergent validity analysis.

| Constructs | Label | Unstd. | S.E. | *t*-Value | *p* | Std. | SMC | CR | AVE | Cronbach's Alpha |
|---|---|---|---|---|---|---|---|---|---|---|
| DT | DT1 | 1.000 | | | | 0.753 | 0.567 | 0.820 | 0.542 | 0.809 |
| | DT2 | 1.763 | 0.112 | 15.808 | *** | 0.940 | 0.884 | | | |
| | DT3 | 1.168 | 0.097 | 12.056 | *** | 0.606 | 0.367 | | | |
| | DT4 | 1.103 | 0.094 | 11.727 | *** | 0.591 | 0.349 | | | |
| ATT | ATT1 | 1.000 | | | | 0.870 | 0.757 | 0.855 | 0.543 | 0.853 |
| | ATT2 | 0.657 | 0.044 | 14.824 | *** | 0.691 | 0.477 | | | |
| | ATT3 | 0.661 | 0.043 | 15.352 | *** | 0.710 | 0.504 | | | |
| | ATT4 | 0.680 | 0.043 | 15.752 | *** | 0.725 | 0.526 | | | |
| | ATT5 | 0.637 | 0.045 | 14.263 | *** | 0.670 | 0.449 | | | |
| SN | SN1 | 1.000 | | | | 0.884 | 0.781 | 0.876 | 0.586 | 0.874 |
| | SN2 | 0.642 | 0.040 | 16.151 | *** | 0.712 | 0.507 | | | |
| | SN3 | 0.650 | 0.040 | 16.442 | *** | 0.721 | 0.520 | | | |
| | SN4 | 0.696 | 0.041 | 16.983 | *** | 0.738 | 0.545 | | | |
| | SN5 | 0.726 | 0.041 | 17.711 | *** | 0.760 | 0.578 | | | |
| PBC | PBC1 | 1.000 | | | | 0.918 | 0.843 | 0.894 | 0.588 | 0.893 |
| | PBC2 | 0.593 | 0.036 | 16.676 | *** | 0.695 | 0.483 | | | |
| | PBC3 | 0.664 | 0.034 | 19.255 | *** | 0.760 | 0.578 | | | |
| | PBC4 | 0.679 | 0.037 | 18.557 | *** | 0.743 | 0.552 | | | |
| | PBC5 | 0.661 | 0.035 | 18.855 | *** | 0.750 | 0.563 | | | |
| | PBC6 | 0.634 | 0.037 | 17.308 | *** | 0.712 | 0.507 | | | |
| INT | INT1 | 1.000 | | | | 0.917 | 0.841 | 0.806 | 0.518 | 0.801 |
| | INT2 | 0.495 | 0.042 | 11.689 | *** | 0.596 | 0.355 | | | |
| | INT3 | 0.554 | 0.045 | 12.370 | *** | 0.630 | 0.397 | | | |
| | INT4 | 0.627 | 0.046 | 13.526 | *** | 0.691 | 0.477 | | | |
| GT | GT1 | 1.000 | | | | 0.895 | 0.801 | 0.902 | 0.607 | 0.879 |
| | GT2 | 0.686 | 0.035 | 19.399 | *** | 0.775 | 0.601 | | | |
| | GT3 | 0.629 | 0.035 | 17.758 | *** | 0.734 | 0.539 | | | |
| | GT4 | 0.693 | 0.037 | 18.867 | *** | 0.762 | 0.581 | | | |
| | GT5 | 0.641 | 0.036 | 17.922 | *** | 0.738 | 0.545 | | | |
| | GT6 | 0.640 | 0.034 | 18.715 | *** | 0.758 | 0.575 | | | |

Significance levels: $p < 0.001$ (***).

Validity includes convergent validity and discriminant validity. Average variance extracted (AVE) reflects the mean of the explanatory power of the latent variables for the observed variables. As can be seen from the results in Table 3, the AVE values of all the constructs are greater than 0.5, which meets the criteria suggested by Fornell and Larcker [107], indicating that the constructs have good convergent validity. Discriminant validity assesses the degree of distinction between the different constructs, and, as suggested by Fornell and Larcker [107], the square root of the AVE value of each construct in the model should be greater than the Pearson correlation coefficient between that construct and the other constructs. The results of the discriminant validity test are given in Table 4, where it can be found that the square roots of the AVE values of all the constructs are greater than the

Pearson correlation coefficient between that construct and the other constructs, indicating that the latent variables in the model have strong discriminant validity.

**Table 4.** Results of the discriminant validity test.

|  | AVE | GT | BI | PBC | SN | ATT | DT |
|---|---|---|---|---|---|---|---|
| GT | 0.607 | 0.779 | | | | | |
| BI | 0.518 | 0.332 | 0.720 | | | | |
| PBC | 0.588 | 0.334 | 0.545 | 0.767 | | | |
| SN | 0.586 | 0.333 | 0.410 | 0.368 | 0.766 | | |
| ATT | 0.543 | 0.312 | 0.512 | 0.359 | 0.338 | 0.737 | |
| DT | 0.542 | 0.383 | 0.480 | 0.335 | 0.379 | 0.383 | 0.736 |

*5.2. Model Fit and Path Coefficients*

To test the goodness of fit of the structural model, this study used AMOS 26.0 software to construct the overall structural relationship model. The 406 sample data obtained from the research questionnaire were used to test the model fit of the theoretical model of the mechanism of DT's impact on the GT of enterprises (as shown in Figure 2). The more ideal the fit index, the closer the model is to the actual situation of the sample [108].

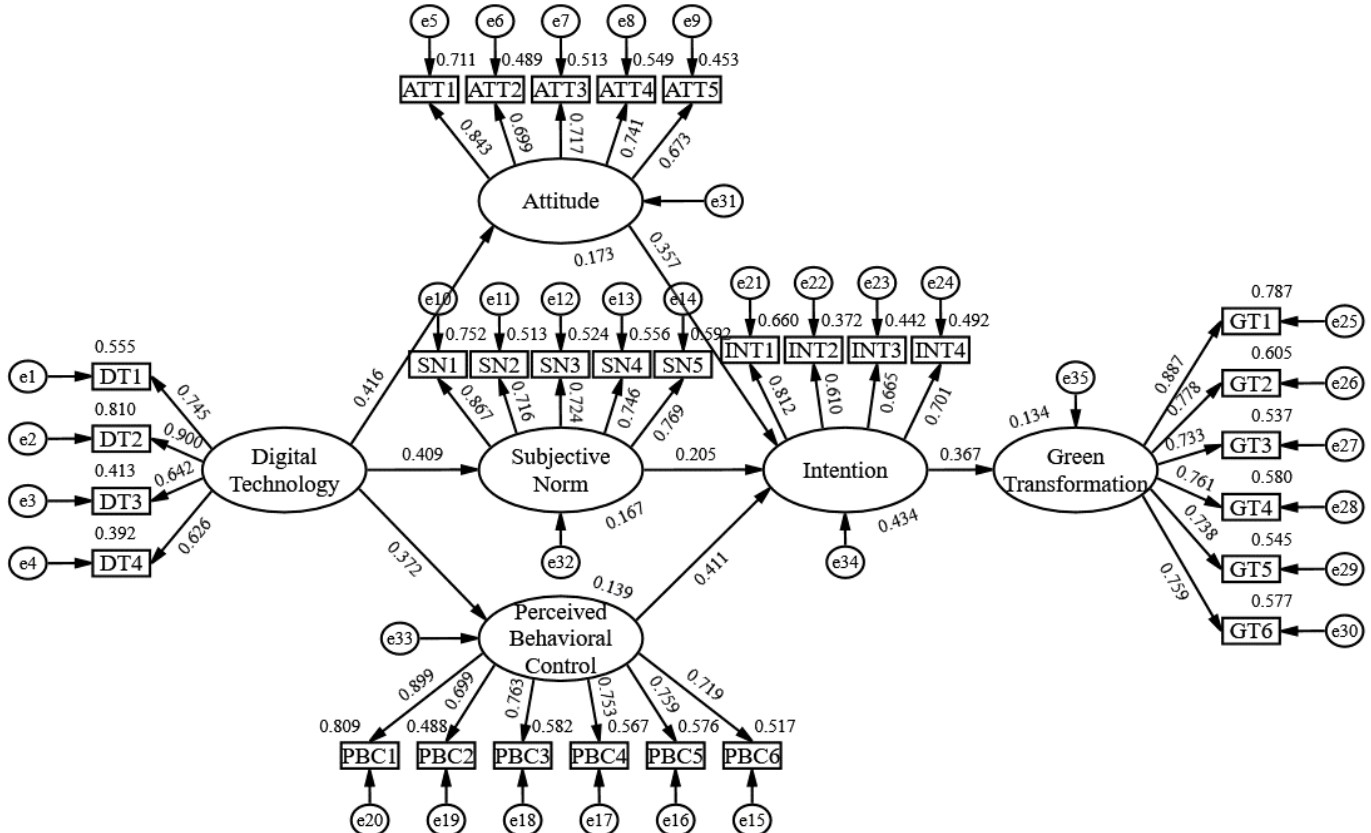

**Figure 2.** Standardized estimation of the theoretical model of the mechanism of DT's impact on the GT of enterprises.

Based on the research findings of Jackson [104], this paper selected Chi-square ($\chi^2$), degrees of freedom (*df*), $\chi^2/df$ ratio, root-mean-square error of approximation (RMSEA), goodness of fit index (GFI), adjusted GFI (AGFI), normed fit index (NFI), Tucker–Lewis index (TLI), and comparative fit index (CFI), a total of nine indicators, to measure the degree of model fit, as shown in Table 5. The model fit indices of the structural model were

all within the acceptable range [99,109–111], which indicated that the structural model had a good fit.

**Table 5.** Model fit indices.

| Fit Indices | $\chi^2$ | *df* | $\chi^2$/*df* | RMSEA | GFI | AGFI | NFI | TLI | CFI |
|---|---|---|---|---|---|---|---|---|---|
| Test value | 814.170 | 398 | 2.046 | 0.051 | 0.881 | 0.861 | 0.879 | 0.928 | 0.934 |
| Recommended values | N/A | N/A | 1~3 | <0.08 | >0.8 | >0.8 | >0.9 | >0.9 | >0.9 |

The results of the path coefficient tests are shown in Table 6. All unstandardized path coefficients passed the significance test of *p* < 0.001, representing a significant impact relationship between the latent variables for all paths in the model. Therefore, H2~H8 were all supported. The above results indicate that the theoretical model of the mechanism of DT's impact on the GT of enterprises has strong predictive and explanatory power.

**Table 6.** Path coefficient of latent variables and hypotheses test.

| Path | Parameter Significance Estimates | | | | Standardized Path Coefficient | Hypothesis | Conclusion |
|---|---|---|---|---|---|---|---|
| | Unstd. | S.E. | C.R. [1] | *p* [2] | | | |
| DT→ATT | 0.805 | 0.112 | 7.223 | *** | 0.416 | H2 | Supported |
| DT→SN | 0.831 | 0.115 | 7.212 | *** | 0.409 | H3 | Supported |
| DT→PBC | 0.772 | 0.115 | 6.697 | *** | 0.372 | H4 | Supported |
| ATT→INT | 0.326 | 0.048 | 6.775 | *** | 0.357 | H5 | Supported |
| SN→INT | 0.179 | 0.044 | 4.096 | *** | 0.205 | H6 | Supported |
| PBC→INT | 0.350 | 0.044 | 8.006 | *** | 0.411 | H7 | Supported |
| INT→GT | 0.453 | 0.070 | 6.466 | *** | 0.367 | H8 | Supported |

[1] C.R. value is the quotient of the unstandardized path coefficient divided by the standard error. When |C.R.| > 1.96, the test result is significant at the 5% level. When |C.R.| > 2.58, the test result is significant at the 1% level. [2] Significance levels: *p* < 0.001 (***).

DT had a significant positive impact on attitude (digitalization drives GT), subjective norms for GT, and perceived behavioral control for GT, all of which trigger the intention to pursue GT. Intention to pursue GT has a significant positive impact on GT. This suggests that when an enterprise is successful in digitalization, it makes a positive contribution to its GT.

*5.3. Mediating Effect*

Mediating effect tests can verify the process and effect of the independent variables' impacts on the dependent variable. Compared to path coefficient tests, mediating effects tests focus more on explaining the "how" and "why" of the impact between variables [112], and therefore, mediating effect tests can often lead to more in-depth findings. The most widely used method for testing mediating effects is the causal step method popularized by Baron and Kenny [113], and Sobel improved on the causal step method with the Sobel Test [114,115]. However, as the Sobel Test requires the assumption that the sampling distribution of mediating effects is normal, an assumption that is often difficult to achieve in practice, it produces biased results [115,116]. Simulation studies have shown that the Bootstrap method tests mediating effects more effectively than the causal step method and the Sobel Test [117,118]. Therefore, this paper instead directly tests the significance of the product of coefficients using the Bootstrap method, which is now generally considered to be better [119]. The Bootstrap method does not require the data to conform to a normal distribution, the standard errors and confidence intervals of the indirect effects are re-estimated through random repeated sampling, and it can be used to test the indirect effects of any mediating variable model [120].

According to the theoretical model of this study, the 'DT' variable drives the 'GT' variable through three pathways. As this is a multiple-mediator model, the test for multiple

mediator effects consists of two components, namely assessing the total indirect effect and the specific indirect effect in the model, as suggested by Holbert [121] and Preacher [122]. Therefore, this paper uses the Bootstrap method to test the mediating effects of the theoretical model with a confidence interval of 95% and a random sample of 5000 times.

The results of the mediating effect tests are shown in Table 7. The z-values for all specific indirect effects and total indirect effects were greater than 1.96, and neither BC nor percentile contained 0 between the minimal and maximal values at the 95% confidence interval, indicating that all specific indirect effects and total indirect effects were significant [116]. Therefore, H9~H11 were all supported. By comparing the mediating effects of the pathways "DT→ATT→INT→GT", "DT→SN→INT→GT", and "DT→PBC→INT→GT", it can be found that the specific indirect effect of "DT→PBC→BI→GT" is the largest, at 0.123, accounting for 39.8% of the total indirect effect. This shows that DT can enhance the capability of enterprises for GT; in other words, it can solve the problem of insufficient motivation and capacity of enterprises for GT, thus enhancing their intention to pursue GT and ultimately promoting the GT of enterprises. This is the most significant pathway for DT to promote the GT of enterprises.

**Table 7.** Mediating effect test.

| Indirect Effect | Point Estimate | Product of Coefficients | | Bootstrapping | | | |
| | | | | BC [3] 95% CI [4] | | Percentile 95% CI | |
| | | SE [1] | Z [2] | Lower | Upper | Lower | Upper |
|---|---|---|---|---|---|---|---|
| DT→ATT→INT→GT | 0.119 | 0.034 | 3.500 | 0.065 | 0.201 | 0.063 | 0.196 |
| DT→SN→INT→GT | 0.067 | 0.027 | 2.481 | 0.026 | 0.131 | 0.025 | 0.129 |
| DT→PBC→INT→GT | 0.123 | 0.037 | 3.324 | 0.064 | 0.212 | 0.062 | 0.209 |
| Total indirect effect | 0.309 | 0.076 | 4.066 | 0.177 | 0.474 | 0.181 | 0.478 |

[1] SE stands for Standard Error. [2] Z is the quotient of the point estimate divided by the standard error. The test is significant at the 5% level when $|Z| > 1.96$. [3] BC stands for bias-corrected. [4] CI stands for confidence interval, and the results were obtained by Bootstrap performed 5000 times.

## 6. Discussion

In the context of the rapid development of the digital economy and the increasing digitalization of enterprises, some scholars have started to focus on the relationship between digitalization and the green development of enterprises, but little literature has explored the impact of DT on GT, especially given the lack of empirical evidence from the micro-scale perspective of enterprises [123]. From a practical point of view, when implementing GT strategies, active digitalization and conscious acceleration of digitalization are required to cross the digital divide and achieve GT more quickly [78]. Based on this, this paper introduces the exogenous variable of DT based on the TPB and uses an SEM approach to empirically test the research model. The aim is to fill the gaps in the existing research by studying the impact relationship between DT and the GT of enterprises and the intermediate transmission mechanism of the impact of DT on the GT of enterprises in China.

This study indicates that DT facilitates the process of GT in enterprises. This finding is in line with Zhang [9] and Waqas [124]. Digital transformation brings productivity and organizational performance improvements to companies, making it an inevitable choice for the future development of Chinese companies [14]. Based on this background, Made in China 2025 plans to make digitalization of the manufacturing industry one of the directions for the transformation and upgrading of Chinese manufacturing enterprises. As China continues to promote the digital transformation of enterprises, some studies have found that digital transformation can also contribute to the improvement of corporate environmental performance [125]. This study further demonstrates that DT can help enterprises achieve GT, and this finding provides an idea to promote a new model of "Digitalization + Greening" synergistic development for China's enterprises to achieve high-quality development [126].

In addition, considering that the theory of planned behavior mainly explains the general decision making of human behavior and the process of occurrence from the perspective of rationality and is not fully controllable by individual will, to further research, this paper conducts a mediation effect test on the multiple mediation model, and the results show that DT drives the GT of enterprises through three paths, which are "DT→ATT→INT→GT", "DT→SN→INT→GT", and "DT→PBC→INT→GT". By comparing the mediating effects of these three paths, it can be found that the specific indirect effect of "DT→PBC→INT→GT" is the largest, which indicates that the degree of operability (capacity, resource elements, etc.) is the most important factor for enterprises to consider when carrying out GT. The digitalization of enterprises can improve resource allocation efficiency [127] and reduce information asymmetry [128], and as the digital transformation progresses, enterprises can rely on DT to make internal adjustments to cope with the changing external environment, thus enhancing dynamic capabilities [129]. At a theoretical level, the rapid development of DT has increased the resource endowment of companies, which in turn has changed the way companies communicate with their stakeholders. Based on the cost effects and empowerment effects, we can find that DT helps firms reduce the cost of transformation, accumulate resources, and improve their learning capabilities and skills. More importantly, companies integrate and reconfigure traditional resources through DT to mitigate the disadvantages brought about by resource exclusivity. In addition, when facing multiple stakeholders, DT companies can accurately capture the needs of different stakeholders and have better responsiveness. In conclusion, DT effectively improves corporate environmental management, and the resources of the company become a key influencing factor in the GT of the enterprise.

Internal and external social pressures have a significant impact on the drive for corporate GT. Based on the focus theory of normative conduct, the environmental behavior of companies is influenced by their own ethical norms as well as by external environmental constraints. The use of DT provides a platform for information dissemination, allowing government and external regulators and potential investors to access corporate information in a timely manner [130], increasing the exposure of corporate environmental violations and effectively reducing under-reporting of pollution. To maintain their own social image, enterprises are motivated to take the initiative to save energy, reduce emissions, and adjust their production inputs in a timely manner, thus forming a positive incentive for them to develop in green ways. In addition, the development of DT has increased peer competition among enterprises, and more and more enterprises are developing environmentally friendly projects to meet consumer demand, raising their awareness of regulation and thus promoting GT.

For behavioral attitudes, the more positive a manager's evaluation of driving GT through digitalization, the greater the likelihood of a behavioral response. Based on the cost effect and empowerment effect, enterprise digitalization is conducive to reducing the cost of energy savings and emission reduction in enterprises, while manufacturing enterprises enhance the efficiency of factor allocation in the production and operation process of enterprise green products and innovation investment through the application of DT to promote GT. With the digital transformation of enterprises, enterprises can use the Internet, big data, artificial intelligence, and other new-generation information technology to access all kinds of information in the process of green innovation. This is conducive to broadening the depth and breadth of information available to enterprises and effectively enhancing the traceability and completeness of information, thereby reducing the cost of the enterprise GT process. DT can be used to effectively integrate information from within and outside the enterprise and realize the transfer, flow, and sharing of information in the green transformation process [131]. In the process of green investment and innovation, enterprises can accurately grasp market supply and demand, effectively identify market opportunities and avoid market risks, reasonably guide the flow of green investment capital, and improve the efficiency of GT factor allocation by deeply mining important market-related information [132].

## 7. Conclusions

This paper examines the non-economic consequences of the digitalization of enterprises from the perspective of GT. The results of the study help enterprises balance digitalization and GT, which is important for the modernization concept of "harmony between man and nature" and the achievement of high-quality economic development in the new era. In addition, this study helps to deepen our micro-level understanding of the complex relationship between digitalization and corporate GT. At the same time, a theoretical model of the mechanism of DT's impact on enterprises' GT is constructed based on the TPB, and an in-depth analysis of the intermediate influence mechanism of DT on the GT of Chinese enterprises helps to clarify the complex theoretical relationship between the two.

This paper empirically proves that enterprises can quickly adapt to the digital economy through internal digitalization while promoting their GT. DT indirectly and positively impacts the intention to pursue GT by impacting the attitude that digitalization drives GT, the subjective norm, and the perceived behavioral control, which ultimately positively impacts actual behavior towards the GT of enterprises. Based on the above research findings, to better exploit the incentive effect of DT on enterprises to carry out GT and achieve green growth, this paper puts forward the following policy recommendations:

Firstly, the government should follow the development trend of the digital economy and accelerate the construction of new infrastructure such as 5G base stations, the Internet of Things, data centers, AI, and blockchain to provide a favorable external environment for the digital transformation of enterprises. At the same time, financial institutions should be encouraged to provide the necessary financial support for enterprises' digital transformation to ease the financial pressure on their digital transformation and accelerate it. In addition, the government can build a sustainable R&D and innovation system based on DT to enable enterprises to achieve environmental governance and green development through digitalization and to give play to the role of DT in driving the GT of enterprises.

Secondly, enterprises should actively digitize and strengthen their digital mindset. Digital literacy should be cultivated and digital skills enhanced among employees, and DT should be applied throughout all processes of enterprise production, operation, and management. When evaluating the value of digital transformation, enterprises should not only look at the direct economic benefits brought by the transformation in terms of cost reduction and efficiency gains but should comprehensively evaluate the full financial and strategic benefits that digital transformation brings to the enterprise in various aspects such as increasing efficiency, reducing costs, promoting innovation and research and development, and improving environmental governance.

Thirdly, enterprises should rely on DT to achieve the sharing of internal and external knowledge and information, stimulate the potential of the production and consumption sides, improve the productivity of enterprises, and promote the construction of a good digital ecosystem. At the same time, enterprises should invest more in green R&D projects and accelerate the development and application of green technologies such as energy savings and emission reduction. From efficiency improvement to technological advancement, enterprises should promote the synergistic development of "Digitalization + Greening" to achieve the goal of green and sustainable development.

Finally, with the rapid development of the digital economy, the environmental protection sector should adopt taxation and financial subsidies to guide green investment in the capital market, encourage enterprises to better invest in environmental protection, and improve their environmental governance capabilities. At the same time, regulators should improve the system of corporate environmental information disclosure to enhance the transparency of corporate information and provide effective external supervision and restraint for corporate GT. The government should encourage and promote representative enterprises to enhance the awareness of GT of surrounding enterprises, thereby enriching the green industrial chain and accelerating the development of a green economy.

Although this paper has conducted a multi-level empirical study, there are still certain limitations, and there is a need to continue to deepen the exploration of the mechanism of

DT on the GT of enterprises. Firstly, the study only focuses on the impact of DT on the GT of enterprises and does not break down the two. In the future, the impact of different types of digitalization on the green cultural transformation, green strategic transformation, and green management transformation of enterprises can be further explored. Secondly, there may be regional asymmetry or industry asymmetry in the mechanism of DT's impact on enterprises' GT. In the future, the scope of the study can be extended to different industries in multiple regions and further global discussions on the effect of DT on enterprise GT to improve the applicability of the theoretical framework. Third, the sample is based on manufacturing companies in different sectors. Further results can be obtained if the data are broken down by type of manufacturing company. In addition, future exploratory analysis of the model could be conducted using more optimized methods, while exploring more accurate GT paths based on normative behavioral focus theory, resource-based view perspectives, etc. Finally, whether there is a synergistic effect between DT and corporate GT also needs to be studied, providing a direction for future research.

**Author Contributions:** Conceptualization, Y.G. and Y.T.; methodology, Y.G.; software, Y.T.; validation, Y.G. and Y.T.; formal analysis, Y.T.; investigation, Y.T.; resources, Y.G.; data curation, Y.G. and Y.T.; writing—original draft preparation, Y.T.; writing—review and editing, Y.G.; visualization, Y.G. and Y.T.; supervision, Y.G.; project administration, Y.G.; funding acquisition, Y.G. All authors have read and agreed to the published version of the manuscript.

**Funding:** This research received no external funding.

**Institutional Review Board Statement:** Not applicable.

**Informed Consent Statement:** Not applicable.

**Data Availability Statement:** Not applicable.

**Acknowledgments:** The authors thank the anonymous reviewers and academic editors for their valuable comments.

**Conflicts of Interest:** The authors declare no conflict of interest.

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
