# Peer review of "A Study on the Mechanism of Digital Technology’s Impact on the Green Transformation of Enterprises: Based on the Theory of Planned Behavior Approach"

_sustainability, doi:10.3390/su151511854_

Round 1

Reviewer 1 Report

This article explores the mechanisms by which digital technology impacts the green transformation of businesses and uses the Theory of Planned Behavior to explain this influence. The research indicates that digital technology makes a positive contribution to the green transformation of businesses and reveals the underlying models and mechanisms. Additionally, the study provides an empirical model to assess the relationship between the intention and behavior of businesses in undertaking green transformation. It includes:

l   How the Theory of Planned Behavior helps understand the impact of digital technology on the green transformation of businesses.

l   Key factors influencing the intention of businesses to engage in green transformation through digital technology.

l   Providing empirical testing of the research model using structural equation modeling.

However, there are areas for improvement:

1.      The relevance of the literature to the research content needs to be strengthened. For example, the literature review could present the content of related studies and literature more clearly. Here is a possible way to write it:

l   The Impact of Digital Transformation on Corporate Environmental Management:

Review relevant studies on the impact of digital transformation on corporate environmental management. In addition to citing the research by Jiahui et al. [66], it should explain and discuss how digital transformation improves corporate environmental management and propose further research agendas. Furthermore, discuss the research by Cialdini et al. [67] on the focus theory of normative behavior and reassess the role of norms in human behavior.

l   The Theoretical Foundation and Empirical Research of Green Transformation:

Review the theoretical foundation and empirical research of green transformation. This part may require the use of meta-analysis. Review the micro-level empirical evidence on environmental and economic performance over the past decade and explore the relationship between environmental and economic performance.

2.      Each hypothesis should be listed in Figure 1 to provide readers with a clear understanding of the research framework and its connections.

3.      The results of testing the mediating effects in the study reveal that the impact of digitalization on driving green transformation occurs through three pathways: "DT ATT INT GT," "DT SN INT GT," and "DT PBC INT GT." Among these, the specific indirect effect of "DT PBC INT GT" is the greatest, indicating that the operational feasibility (capabilities and resources) of businesses is a key factor in achieving digitalization-driven green transformation.

However, further in-depth exploration is needed to enhance the contribution of this article.

4.      It is suggested that the organization of sections 6, 7, and 8 be reconsidered by the author. In a journal article, it may not be necessary to explicitly mention future research; it could be implicitly included in the conclusion. Additionally, the conclusion serves as a summary of the article, and there may not be any further content following it.

Reviewer 2 Report

DT and GT are current themes that still generate doubts about their implementation. This research addresses important issues but needs to be better formatted. I leave some suggestions below:

How was the Theoretical Model of section 3.2 constructed? The theoretical model is already a result but is presented before the methodology section. This theoretical model should appear at the beginning of the Results section.

The Methodology section needs to be thoroughly revised. The authors include a Theoretical Basis in the methodology, which should be presented in the Literature review section. It is not presented how the research was conducted; it also needs to be shown how the literature review was done to arrive at the Research Model.

In the results section, you could think about presenting the findings in a way that is easier to be understood managers and decision-makers, such as a table. You can see the following article as a basis, which presents a table showing the positive and negative impacts of the implementation of DT, as well as barriers and opportunities. (https://doi.org/10.3390/su15118562).

Reviewer 3 Report

I wish to thank for the opportunity to review this interesting paper. It deals with an important and highly topical theme, i.e. how digital technology (DT) enhances achievement of corporate green transformation (GT).  The paper constructs a theoretical model based on the theory of planned behavior (TPB), and empirically tests the model using a structural equation model (SEM). The analysis of data from 406 manufacturing firms in China shows that DT has a positive contribution to corporate GT.

Although I am not an expert of the methodologies the authors use, the methods are logically explained, and the argumentation of the authors seems very convincing. The authors explain their model and all its elements in detail admirably. The results are explained in detail.

I think the paper makes a nice contribution to the journal. However, the authors could discuss more how their research contributes these themes globally. For example, European Union makes the same link between digitalization and green transformation in its policies, aiming at synergistic development of digitalization and greening.  I also suggest adding references related to European discussion.

Minor issues:

-         the authors use both digitization and digitalization without defining these concepts, for example: “the pace of enterprise digitization has accelerated,… The digitalization of enterprises is essentially a dynamic…” . Is there a difference? If not, could the authors make a choice and use only one concept. In any case, I suggest adding definitions.

The English language is good, and I only suggest spell check.

Reviewer 4 Report

1- The research claims to test the link of DT on Green transformation by the Chinese firms through mediating links of attitude, subjective norm and PBC. However The use of TPB and the strategic nature of constructs are misaligned, moreover there are flaws in methodology and sample selection process so in my opinion, the paper needs a major reconsideration and rework and should be rejected in present form. 

2-

The proposed model is not grounded in solid theoretical framework. The latent variables of Att, SN and PBC represent enterprise behavior and collective nature/strategic perspective but the methodology lacks any details of how the collective nature of data collection was managed. Who participated in data collection and the firm size/firm characteritstics not clear. 

3-

The utilized items most oftenly represent the environmental readiness and environmental response of the organization hence mere naming of these items with digital technology and green transformation might not be enough to justify the used constructs. The used constructs should be theoretically built and discussed first.

4-

The methodology does not depict how these manufacturing companies were chosen. The random selection of companies is mentioned only without succinct details of company selection/sector selection etc. The Chinese manufacturing sector represents a giant array of small, medium and large enterprises so researchers should devise and clearly state how they approached the sector and which firms were counted in.

5-

H1 The proposition of model fitness is traditionally tested against another model and the results are drawn after model comparison with another model comparison but the researchers did not follow the method prescribed in literature. Hence they should review that for the literature provided here 

https://www.tandfonline.com/doi/full/10.1080/13527266.2021.1881807 and http://dx.doi.org/10.3390/su10030673

6-

The operationalization of various resources have been allocated to SN/PBC without much literature support and backing hence the results and implication drawn might be irrelevant here.  see for example the specific indirect effect of “DT→ PBC→INT→GT” is the largest which indicates that the degree of operability (capacity and resource elements, etc.) is the most important factor for enterprises to consider when carrying out GT. How capacity and resource elements were covered under the construct of 'perceived behavioral control' construct. I would advise the researchers to drop TPB perspective here and adopt a strategic management/RBV perspective or Intellectual capital perspective for building their entire storyline and hypotheses. 

The manuscript needs some revision for proofreading and grammatical error correction by a professional tool.

Reviewer 5 Report

1. introduction must be improved as it lacks support from recent findings and is not explained clearly. the context is vaguely described and the justifications for the conduct of the study are missing. 

2. the authors must provide information regarding their contributions to the current understanding in the introduction section. 

3. the literature lacks coherence and is not supported by relevant references. this includes both theoretical findings and reports.

4. the literature/hypotheses sections are not adequately described. 
i recommend authors tend to each hypothesis separately and present it after its relevant arguments. 

5. the literature review section also needs to be improved regarding its relevance both regarding contextual and theoretical aspects. relationships under examination should be clearly argued upon using the most recent/most relevant findings in the literature (which are abundant) 

6.  which method was used for sampling? 

7. how was the sample size calculated?

8. how was the criteria for data collection processed? 

9. the analytical technique should be justified. considering the measurement model, why PLS SEM was not used (i.e. Hair et al., 2017, 2019). 

10. authors must provide "discussion" referring to each hypothesis based on their findings in the structural model assessment. 

11. conclusions should be revised upon implementing the aforementioned issues. 

12. implications both theoretical and practical should be presented in a specific section where authors highlight their contributions to the field of research and practice. 

13. Limitations should consider theoretical, contextual, methodical, analytical, and other constraining factors. 

the language of the paper is incoherent and contains various errors (e.g. punctuation, sentence structure, wording, and grammatical issues). 

professional editorial services can  be useful for authors. 

Round 2

Reviewer 2 Report

Thanks for answer all my appointments

Author Response

Thank you again for your positive comments and valuable suggestions to improve the quality of our manuscript. And we hope the revised manuscript could be acceptable for you.

Reviewer 5 Report

kudos to authors for improvements in the revised version.

the hypotheses are still not well presented, 

i repeat, the hypothesis should come after their description/explanation by the literature. 

literature specific to the hypothesis -> hypothesis 

while various issues have been resolved, minor editing is needed as there are still grammatical, lexical, and structural issues. 
